# MiVE: Multiscale Vision-language features for reference-guided video Editing

**Tong Wang** [* 1] **Meng Zou** [* 1 2] **Chengjing Wu** [1] **Xiaochao Qu** [1] **Luoqi Liu** [1] **Xiaolin Hu** [3] **Ting Liu** [1]

## Abstract

Reference-guided video editing takes a source video, a text instruction, and a reference image as inputs, requiring the model to faithfully apply the instructed edits while preserving original motion and unedited content. Existing methods fall into two paradigms, each with inherent limitations: decoupled encoders suffer from modality gaps when processing instructions and visual content independently, while unified vision-language encoders lose fine-grained spatial details by relying solely on final-layer representations. We observe that VLM layers encode complementary information hierarchically—early layers capture localized spatial details essential for precise editing, while deeper layers encode global semantics for instruction comprehension. Building on this insight, we present **MiVE** (**M**ult**i**scale **V**ision-language features for reference-guided video **E**diting), a framework that repurposes VLMs as multiscale feature extractors. MiVE extracts hierarchical features from Qwen3-VL and integrates them into a unified self-attention Diffusion Transformer, eliminating the modality mismatch inherent in cross-attention designs. Experiments demonstrate that MiVE achieves state-of-the-art performance by ranking highest in human preference, outperforming both academic methods and commercial systems.

## 1. Introduction

Reference-guided video editing aims to propagate edits from a reference image—typically the first frame modified to re-

[1]MT Lab, Meitu Inc., Beijing 100083, China [2]Beijing University of Posts and Telecommunications, Beijing 100876, China [3]Department of Computer Science and Technology, BNRist, IDG/McGovern Institute for Brain Research, Tsinghua University, Beijing 100084, China. Correspondence to: Xiaolin Hu <xlhu@tsinghua.edu.cn>, Ting Liu <lt@meitu.com>.

*Proceedings of the 43rd International Conference on Machine Learning*, Seoul, South Korea. PMLR 306, 2026. Copyright 2026 by the author(s).

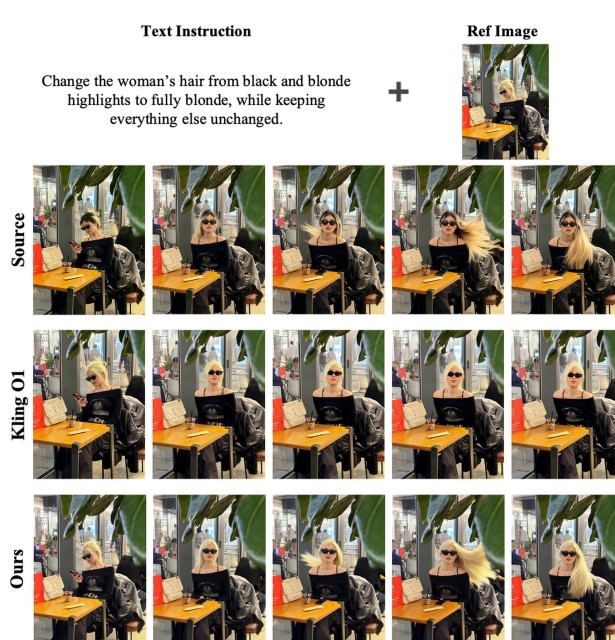

**Text Instruction**

Change the woman's hair from black and blonde highlights to fully blonde, while keeping everything else unchanged.

**+**

**Ref Image**

Source

Kling O1

Ours

*Figure 1.* Qualitative comparison on reference-guided video editing. MiVE faithfully propagates edits from the reference image while preserving fine-grained details, outperforming the commercial system Kling O1. See Section 6 for more results.

flect desired changes—throughout an entire video sequence while preserving original motion and unedited content. Formally, given a source video $x_{src}$ and a text instruction $x_{text}$, we first obtain an edited reference image $x_{ref}$ by applying an external image editing model (e.g., FLUX.1 Kontext (Labs et al., 2025)) to the first frame of $x_{src}$ according to $x_{text}$. The objective is to generate an edited video $\hat{x}_{tgt}$ that (1) faithfully propagates the visual attributes and appearance changes specified in $x_{ref}$ across all frames, and (2) maintains the motion dynamics and semantic integrity of $x_{src}$ in regions not intended for modification. As illustrated in Figure 1 and the online Supplementary Videos[1], the hair color in this case should be altered to match the reference image, while the hair style of the original video must also be preserved.

This task presents a fundamental challenge: the model must simultaneously discern *what to modify* from the reference

---

[1]Supplementary Videos and additional visual comparisons are available at https://mivepaper.github.io.

and *what to preserve* from the source. These objectives are inherently competing. While the model must faithfully propagate visual attributes from the reference image, it must also maintain the motion dynamics and semantic integrity of the original video, even for regions occluded in the reference.

Resolving this tension demands deep cross-modal reasoning over the editing instruction and source video. However, existing approaches often encounter fundamental limitations in multimodal fusion. Predominant models adopt *decoupled architectures*, where language models (e.g., T5 (Raffel et al., 2020)) and visual encoders (e.g., SigLIP (Zhai et al., 2023)) process inputs independently (Cheng et al., 2025; Yang et al., 2025a; Li et al., 2025). This separation restricts cross-modal reasoning, especially in reference-guided video editing, as late-stage cross-attention often fails to bridge the semantic gap between modalities.

A natural remedy is to employ Vision-Language Models (VLMs) as unified encoders, as adopted by recent works such as Kling O1 (Chen et al., 2025). However, two critical challenges remain. First, these methods rely solely on final-layer representations, missing fine-grained spatial details essential for video editing (See Supplementary Videos[1] for examples). While hierarchical encoding is well-known in deep neural networks, our analysis confirms VLMs follow this pattern—early layers encode spatial details, final layers capture semantics—motivating multiscale extraction. Second, *how* to inject these features is non-trivial. Cross-attention is asymmetric: visual tokens query conditional features, but conditional tokens remain agnostic to the visual content, limiting fine-grained correspondence. Effective editing demands both richer representations and *bidirectional* fusion within a shared space.

Motivated by this insight, we propose **MiVE** (**Mi**ultiscale **V**ision-language features for reference-guided video **E**diting). MiVE comprises three core components designed to leverage this hierarchical complementarity. First, a *multiscale context extraction* module harvests features from both early and final VLM layers, projecting them into the diffusion latent space. Second, *reference-aware latent encoding* concatenates reference frames with video latents to preserve structural cues. Third, a *unified self-attention backbone* processes all tokens—visual latents and multi-level VLM features—within a single attention manifold. Unlike conventional cross-attention, our unified token modeling enables the model to learn modality alignment within a shared space, facilitating deep integration of fine-grained spatial details and global semantic context throughout the generation process.

Extensive experiments (Section 6.2; see Supplementary Videos[1] for visual comparisons) across diverse editing scenarios demonstrate that MiVE achieves state-of-the-art performance, ranking first in human evaluations and surpassing both leading academic methods and commercial systems in automated evaluation using Gemini-3-Flash (Google, 2026) across six dimensions. Systematic ablation studies (Sec. 6.3) further validate our design choices, confirming that integrating multiscale VLM features via unified self-attention is essential for balancing semantic understanding with spatial fidelity.

## 2. Related Work

### 2.1. Conditioning Encoders

Most multimodal video models follow a decoupled architecture, where T5-like (Raffel et al., 2020) language models encode text instructions and CLIP-style visual encoders process reference images (Wang et al., 2025a; Yang et al., 2025b). This separation introduces a semantic gap that cross-attention cannot fully bridge. Vision-Language Models (VLMs) offer a unified alternative by jointly processing visual and textual inputs. Recent VLMs such as Qwen3-VL (Bai et al., 2025b) and MiniCPM-V-4.5 (Yu et al., 2025) demonstrate that joint encoding enables richer cross-modal interactions, enabling deeper cross-modal integration for video editing tasks.

### 2.2. Video Editing

Existing video editing techniques can be broadly categorized into mask-guided methods and mask-free methods.

**Mask-guided editing.** Recent work such as VACE (Jiang et al., 2025) and VideoPainter (Bian et al., 2025) employs explicit segmentation masks for spatial control in video editing. VACE proposes a unified framework that integrates various visual conditions and optionally incorporates a reference frame as guidance through latent-space concatenation of noisy latents, mask latents, and conditioning video. VideoPainter adopts a different approach with a dual-branch Diffusion Transformer that requires an edited first frame to anchor identity features, which are subsequently propagated to masked regions via an identity resampling strategy. While effective in controlled settings, these methods encounter difficulties when object motion is rapid or backgrounds are complex—obtaining accurate, temporally consistent masks becomes non-trivial and can limit their practical applicability.

**Mask-free editing.** Recent efforts have moved toward end-to-end editing paradigms to reduce reliance on explicit masks. Lucy Edit (Team, 2025) establishes a foundational model through channel concatenation but struggles with fine-grained local manipulation. Wan-Animate (Cheng et al., 2025) incorporates reference images and control signals (face keypoints and pose skeletons) for character replacement and animation; however, it cannot adequately handle environment relighting or occlusion recovery.

Several methods infer editing regions implicitly from text instructions. LoVoRA (Xiao et al., 2025) learns object boundaries across space and time but still requires mask supervision during training. VideoCoF (Yang et al., 2025a) introduces Chain-of-Frames reasoning but incurs prohibitive computational costs for longer videos. ReCo (Zhang et al., 2025) reframes the problem as contextual learning, though this approach also introduces significant overhead.

Other methods explore implicit prior injection: Ditto (Bai et al., 2025a) introduces a context branch, while ICVE (Liao et al., 2025) encodes priors into tokens that participate in DiT attention. Despite their architectural efficiency, these methods often struggle with complex scenes and fine-grained details, motivating our unified self-attention approach with multiscale VLM features.

## 3. Motivation

Recent image editing models leverage deep VLM representations and explicit semantic reasoning (Wu et al., 2025; Google, 2026), achieving strong instruction following. However, extending this to video editing faces an additional challenge: strict temporal consistency is required. Existing VLM-based methods rely solely on final-layer representations, which, as noted in Section 1, sacrifice fine-grained spatial details essential for faithful editing. To validate this hypothesis and guide our design, we investigate attention patterns across VLM layers via a diagnostic framework, uncovering a hierarchical shift that motivates our multiscale extraction strategy.

### 3.1. Diagnostic Framework

To analyze how VLMs align text with video, we extract cross-modal attention dynamics from each Transformer layer. First, the input video is projected into visual tokens $\mathbf{B} \in \mathbb{R}^{M \times d}$ and concatenated with text embeddings $\mathbf{E} \in \mathbb{R}^{N \times d}$ to form a unified context $\mathbf{C} = [\mathbf{E}; \mathbf{B}]$, where $N$, $M$, and $d$ denote the number of text tokens, video tokens, and hidden dimension, respectively. The full attention matrix $\bar{\mathbf{A}}^{(l)}$ at layer $l$ can be decomposed as:

$$\bar{\mathbf{A}}^{(l)} = \begin{bmatrix} \mathbf{E} \\ \mathbf{B} \end{bmatrix} \begin{bmatrix} \mathbf{E}^\top & \mathbf{B}^\top \end{bmatrix} = \begin{bmatrix} \mathbf{E}\mathbf{E}^\top & \mathbf{E}\mathbf{B}^\top \\ \mathbf{B}\mathbf{E}^\top & \mathbf{B}\mathbf{B}^\top \end{bmatrix}. \quad (1)$$

The top-right block $\mathbf{E}\mathbf{B}^\top$ captures how text grounds onto video, which we denote as the Cross-Modal Diagnostic Matrix $\mathbf{A}_{\text{txt}\to\text{vis}}^{(l)} = \mathbf{E}\mathbf{B}^\top \in \mathbb{R}^{N \times M}$(see Appendix A for details).

For visualization , we compute the mean attention across the text dimension and reshape to restore the spatio-temporal structure.

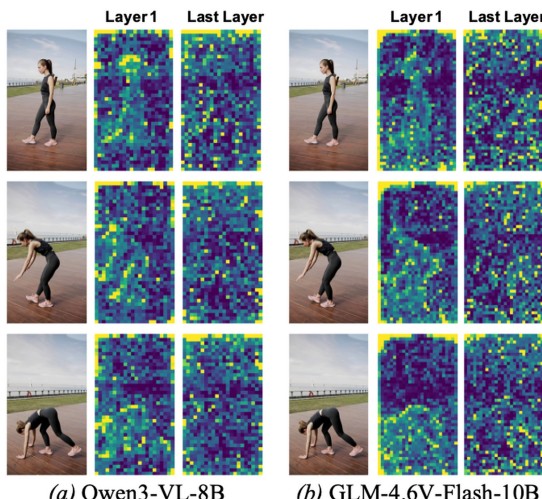

*(a)* Qwen3-VL-8B      *(b)* GLM-4.6V-Flash-10B

*Figure 2.* Cross-modal attention visualization via Section 3.1. Maps represent $\mathbf{A}_{\text{txt}\to\text{vis}}^{(l)} = \mathbf{E}\mathbf{B}^\top$ (**E**: text features, **B**: visual tokens). Layer 1 precisely localizes the human silhouette, while the final layer exhibits diffuse global patterns.

*Table 1.* Attention Mask Ratio ($R_{\text{mask}}$) at different relative depths ($d = l/L$). $d = 0$ denotes the first layer, while $d = 1.0$ denotes the final layer. Higher values indicate stronger spatial localization.

| Model | $d = 0$ | 1/6 | 1/3 | 1/2 | 2/3 | 5/6 | 1.0 |
|---|---|---|---|---|---|---|---|
| Qwen3-VL-8B | 0.366 | 0.296 | 0.253 | 0.197 | 0.226 | 0.177 | 0.228 |
| GLM-4.6V-10B | 0.333 | 0.243 | 0.264 | 0.256 | 0.301 | 0.290 | 0.270 |

### 3.2. Cross-Modal Attention Analysis

We validate our diagnostic framework by evaluating Qwen3-VL-8B (36 layers) and GLM-4.6V-Flash-10B (40 layers) on 100 human-centric video sequences (2–8 seconds, 720P). Let $\mathbf{E}$ denote the human-related text features for localizing the target person, obtained via the text prompt "the person in the video," and let $\mathbf{B}$ denote the visual features extracted from the video. The Cross modal diagnostic matrix $\mathbf{A}_{\text{txt}\to\text{vis}}^{(l)} = \mathbf{E}\mathbf{B}^\top$ captures how the model grounds textual human semantics onto visual representations to localize the target person at layer $l$. Qualitatively, As shown in Figure 2, the first layer precisely outlines the human silhouette, while the final layer exhibits nearly uniform attention. To quantify this, we employ SAM2 (Ravi et al., 2025) to generate human-centric masks $\mathbf{M}$ and define the Attention Mask Ratio:

$$R_{\text{mask}}^{(l)} = \frac{\sum_{i=1}^{N} \sum_{j \in \Omega} \mathbf{A}_{\text{txt}\to\text{vis}}^{(l)}[i,j]}{\sum_{i=1}^{N} \sum_{j=1}^{M} \mathbf{A}_{\text{txt}\to\text{vis}}^{(l)}[i,j]}, \quad (2)$$

where $\Omega$ denotes visual token indices within $\mathbf{M}$. As shown in Table 1, $R_{\text{mask}}$ is higher in early layers and lower in deeper layers, confirming that early layers capture spatial details while final layers encode holistic semantics. Moreover, Qwen3-VL exhibits stronger spatial localization in

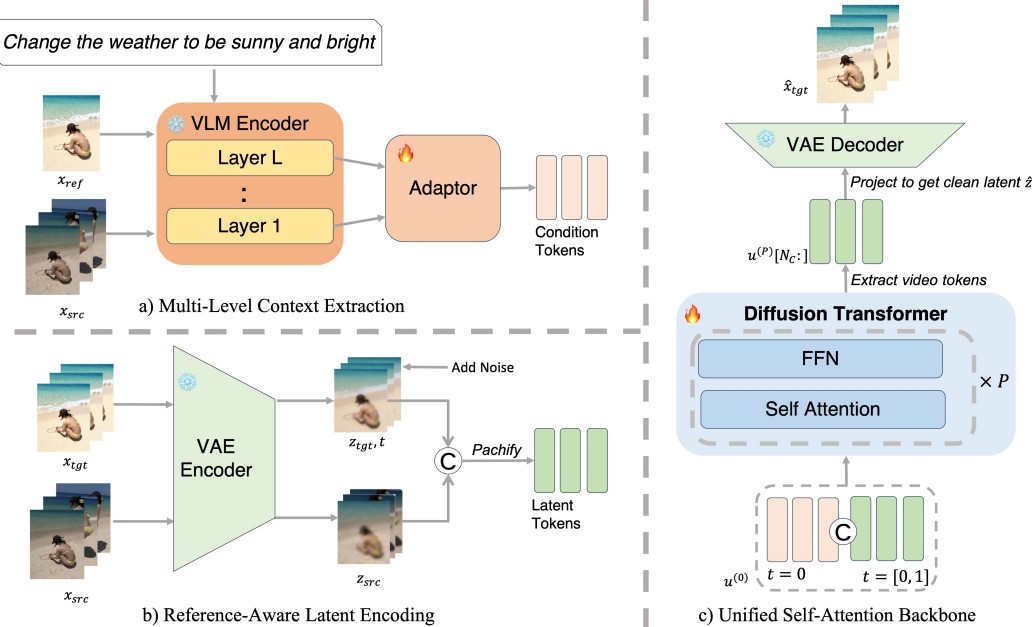

*Figure 3.* **Overview of MiVE.** (a) Multi-level features from Qwen3-VL's first and last layers are projected to condition tokens **c**. (b) Target and source videos are VAE-encoded; the reference latent is prepended temporally, then two branches are concatenated along channels. (c) Condition and latent tokens are jointly processed by DiT blocks with per-token adaptive modulation, where stationary tokens (condition + reference) use fixed time embeddings. During training, the VLM encoder and VAE are frozen and only the adapters and DiT blocks are optimized; during inference, the target latent is initialized from pure noise.

early layers ($R_{\text{mask}} \approx 0.37$ vs. $0.33$), making it more suitable for extracting complementary multi-scale features. We therefore adopt Qwen3-VL as the backbone for our method.

## 4. Method

### 4.1. Overall Architecture

As illustrated in Figure 3, our approach comprises three components: (i) a *Multi-Level Context Extraction* module that extracts conditioning tokens from multi-modal instructions; (ii) a *Reference-Aware Latent Encoding* module that encodes visual content into latent tokens via the VAE encoder; and (iii) a *Unified Self-Attention Backbone* that concatenates VLM and VAE tokens within a shared representation space and denoises through a diffusion transformer (DiT). The model is trained using the flow matching objective (Lipman et al., 2023). After denoising, video tokens are decoded via the VAE decoder to produce the final video.

### 4.2. Multi-Level Context Extraction

Motivated by the observation that different VLM layers encode complementary information (Section 3.2), we design a multi-level feature extraction strategy using Qwen3-VL-8B (Bai et al., 2025b) as the unified encoder for the textual instruction $\mathbf{x}_{\text{text}}$, reference image $\mathbf{x}_{\text{ref}}$, and source video $\mathbf{x}_{\text{src}}$. As shown in Figure 3a, To leverage both fine-grained

structural representations and holistic scene context, we extract features from the first and last layer hidden states of the VLM, denoted as $\phi_1(X), \phi_L(X) \in \mathbb{R}^{S \times D_{\text{VLM}}}$, where $X = \{\mathbf{x}_{\text{text}}, \mathbf{x}_{\text{ref}}, \mathbf{x}_{\text{src}}\}$ is the multi-modal input set, and $S$ and $D_{\text{VLM}}$ denote the sequence length and hidden dimension of the VLM, respectively.

We introduce a lightweight adapter that processes VLM features from each selected layer through independent branches to align them with the DiT latent space. Specifically, for each layer $i \in \{1, L\}$, the feature $\phi_i$ is transformed into a projected representation $\tilde{\phi}_i$ via RMSNorm followed by a linear projection:

$$\tilde{\phi}_i = \text{Linear}_i\left(\text{RMSNorm}(\phi_i)\right) \in \mathbb{R}^{S \times \frac{D}{2}}, \qquad (3)$$

where $D$ is the hidden dimension of the DiT backbone. These multi-level features are then concatenated along the feature dimension to obtain the raw conditioning representation:

$$\mathbf{c}_{\text{raw}} = \text{Concat}_D\left(\tilde{\phi}_1, \tilde{\phi}_L\right) \in \mathbb{R}^{S \times D}. \qquad (4)$$

Finally, $\mathbf{c}_{\text{raw}}$ is processed through a fusion layer to produce the refined conditional tokens **c**:

$$\mathbf{c} = \text{Linear}_{\text{fuse}}(\mathbf{c}_{\text{raw}}) \in \mathbb{R}^{N_c \times D}, \qquad (5)$$

where $N_c$ denotes the number of conditional tokens. Notably, the conditional tokens **c** remain invariant to the diffusion timestep $t$, in which we fix the modulation at $t = 0$,

thereby serving as a stationary semantic anchor that guides the DiT backbone throughout the denoising process.

### 4.3. Reference-Aware Latent Encoding

To preserve the structural integrity of the source video while anchoring the generation to the reference appearance, as illustrated in Figure 3b, we encode all visual inputs into a shared latent space via a pre-trained VAE. Let $\mathcal{E}$ and $\mathcal{D}$ denote the VAE encoder and decoder, respectively. Given an input of spatial resolution $H \times W$ and $T$ frames, the encoder produces latents in $\mathbb{R}^{T' \times C \times H' \times W'}$, where $T' = \lfloor T/4 \rfloor + 1$, $H' = H/8$, and $W' = W/8$.

**Training.** Given paired training data (Section 5.1), each train sample consists of a source video $\mathbf{x}_{\text{src}}$, a target video $\mathbf{x}_{\text{tgt}}$, and the reference image $\mathbf{x}_{\text{ref}}$ (the first frame of $\mathbf{x}_{\text{tgt}}$). We encode them into latent representations $\mathbf{z}_{\text{src}} = \mathcal{E}(\mathbf{x}_{\text{src}})$, $\mathbf{z}_{\text{tgt}} = \mathcal{E}(\mathbf{x}_{\text{tgt}})$, and $\mathbf{z}_{\text{ref}} = \mathcal{E}(\mathbf{x}_{\text{ref}})$.

At diffusion timestep $t$, let $\tilde{\mathbf{z}}_t$ denote the noisy version of $\mathbf{z}_{\text{tgt}}$. We construct the DiT input by concatenating two temporally-augmented branches along the channel dimension:

$$
\mathbf{z}_t = \text{Concat}_C \left( \underbrace{[\mathbf{z}_{\text{ref}}; \tilde{\mathbf{z}}_t]}_{\text{noisy target}}, \underbrace{[\mathbf{z}_{\text{ref}}; \mathbf{z}_{\text{src}}]}_{\text{control}} \right) \in \mathbb{R}^{(T'+1) \times 2C \times H' \times W'}
\tag{6}
$$

where $\text{Concat}_C$ denotes channel-wise concatenation. By prepending the clean reference latent $\mathbf{z}_{\text{ref}}$ to both branches, the model maintains a consistent appearance anchor throughout the denoising process.

**Inference.** During inference, only the source video $\mathbf{x}_{\text{src}}$ and an edited reference image $\mathbf{x}_{\text{ref}}$ (obtained via an external image editor) are available. We sample initial noise $\tilde{\mathbf{z}}_T \sim \mathcal{N}(0, \mathbf{I})$ and construct $\mathbf{z}_T$ following Eq. (6).

In both training and inference, the joint latent $\mathbf{z}_t$ is mapped into visual tokens $\mathbf{v} \in \mathbb{R}^{N_v \times D}$ via a patch embedding layer, where $N_v = (T'+1) \cdot H' \cdot W'/p^2$ denotes the number of spatio-temporal patches with spatial patch size $p$.

### 4.4. Unified Self-Attention Backbone

Unlike decoupled architectures that fuse modalities through separate cross-attention branches, our backbone processes both conditional context and visual latent tokens within a unified self-attention manifold (Figure 3c). We concatenate the conditional tokens $\mathbf{c}$ (Section 4.2) with $\mathbf{v}$ to form the unified input:

$$
\mathbf{u}^{(0)} = [\mathbf{c}; \mathbf{v}] \in \mathbb{R}^{(N_c + N_v) \times D},
\tag{7}
$$

where $\mathbf{u}^{(0)}$ represents layer 0 input of the DiT. This unified sequence is processed through $P$ Diffusion Transformer

blocks. Following Z-Image (Team et al., 2025), we apply per-token adaptive layer normalization (AdaLN) to differentiate *clean* tokens (context $\mathbf{c}$ and reference frame patches) from *noisy* tokens (remaining video patches). Clean tokens are modulated as fully denoised ($t=0$), while noisy tokens are modulated according to the current timestep $t \in [0, 1]$. This ensures the reference frame remains a stable appearance anchor throughout denoising. After $P$ blocks, we extract the visual tokens and project them through an output head:

$$
\hat{\mathbf{z}}_0 = \text{Unpatchify}\big(\text{Head}(\mathbf{u}^{(P)}[N_c:])\big) \in \mathbb{R}^{(T'+1) \times C \times H' \times W'}
\tag{8}
$$

where $[N_c:]$ denotes slicing from index $N_c$ onward, i.e., extracting visual tokens while discarding the conditional tokens. The final edited video is recovered by discarding the reference frame and decoding: $\hat{\mathbf{x}}_{\text{tgt}} = \mathcal{D}(\hat{\mathbf{z}}_0[1:])$.

## 5. Datasets and Benchmark

### 5.1. Training Data Construction

As mentioned in 4, our model relies on paired data in training. We construct a training dataset of approximately 30K video editing pairs from two complementary sources. All samples undergo quality filtering using Qwen3-VL-8B, which evaluates aesthetic quality, instruction-video alignment, and temporal consistency (see Appendix B for details).

**(1) OpenVE-3M.** OpenVE-3M (He et al., 2025) is an open-source paired video editing dataset covering six categories: subtitle editing, global style transfer, local object addition, background change, local object removal, and local modification. Through visual inspection, we verified that samples scoring above 9.3 consistently meet human quality standards; we thus retain only such samples and cap each category at 4,000 pairs due to computational constraints, yielding 24K pairs in total.

**(2) Synthetic Human-Centric Data.** To enhance human-centric editing, we develop a synthetic pipeline that extracts human foregrounds via segmentation and composites them with diverse background videos ($B$), producing three editing types: (i) object deletion: $V_{\text{comp}} \rightarrow B$; (ii) object addition: $B \rightarrow V_{\text{comp}}$; (iii) background replacement: $V_{\text{comp}} \rightarrow V_{\text{orig}}$. Qwen3-VL generates paired captions and editing instructions. After filtering, this source contributes 6K pairs.

### 5.2. Evaluation Benchmark

We evaluate on a curated benchmark of 60 videos at 720P resolution, divided into two subsets based on editing complexity:

*1) Simple Subset (Localized Editing):* 30 cases targeting localized modifications, sourced from: (i) RoseBench (Miao

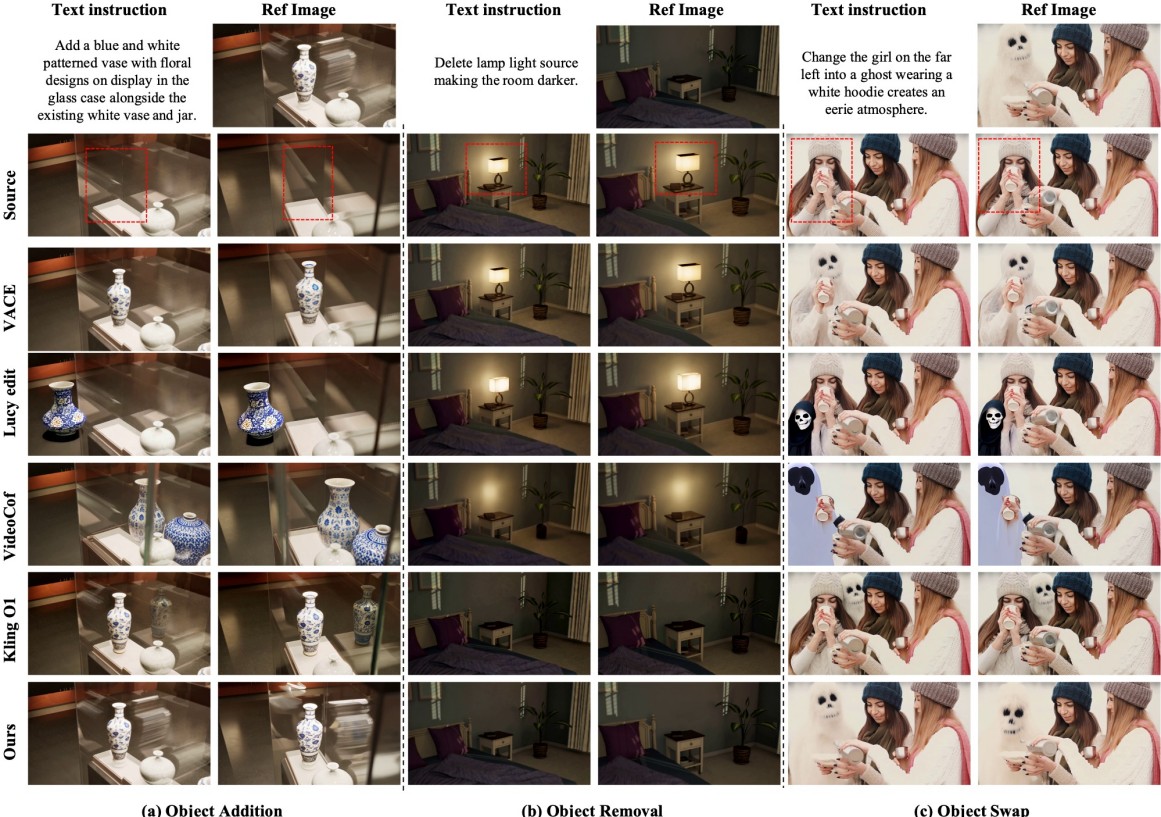

(a) Object Addition  (b) Object Removal  (c) Object Swap

*Figure 4.* **Qualitative comparison on the simple-scenario benchmark.** In simple scenarios, our model accurately captures localized modifications and environmental cues like shadows and reflections. See Supplementary Videos[1] for details.

et al., 2025)—10 sequences for object deletion and 10 for object addition (by swapping source-target pairs); (ii) VP-Bench (Bian et al., 2025)—10 sequences from the "Edit" split, focusing on fine-grained texture and color changes.

*2) Complex Subset (Holistic Editing):* 30 high-quality portrait videos involving holistic scene transformations, including atmosphere transfer, lighting redistribution, background replacement, and complex scene transitions. Crucially, these sequences lack explicit editing masks, requiring the model to perform semantic reasoning and maintain global consistency autonomously.

Due to potential licensing restrictions on some source videos, we are unable to publicly release the complete benchmark. However, we will make the benchmark available to researchers upon reasonable request for non-commercial research purposes.

## 6. Experiments

### 6.1. Implementation Details

Our model is initialized from Wan2.1-T2V-14B (Wang et al., 2025a) self attention blocks and fine-tuned at 720P resolu-

tion with 81 frames. We use AdamW with learning rate $3 \times 10^{-5}$, $\beta = (0.9, 0.999)$, and $\epsilon = 10^{-8}$. Training is conducted on 8 NVIDIA H100 GPUs for 8,000 steps ($\sim$2 epochs) with batch size 1 per GPU, taking approximately 65 hours in total. We apply 200 warmup steps and gradient clipping at 1.0. At inference, generating an 81-frame 720P clip on a single H100 takes $\sim$6.5 min end-to-end (Qwen3-VL forward $\sim$3 s, DiT denoising $\sim$328 s, VAE decoding $\sim$35 s) with peak GPU memory of 50 GB.

### 6.2. Comparison with State-of-the-Art Methods

**Baselines Methods.** We compare against VACE (Jiang et al., 2025), LucyEdit (Team, 2025), VideoCof (Yang et al., 2025a), WanAnimate (Cheng et al., 2025), and commercial system Kling O1 (Chen et al., 2025). As shown in Table 2, methods are categorized by whether they require a reference image (Ref.). Since VACE additionally requires precise spatial masks, it is evaluated only on the Simple Subset. WanAnimate is excluded from the Simple Subset as it exclusively supports portrait video editing and cannot handle non-portrait data.

**Evaluation Protocols.** We adopt a dual evaluation strategy: (1) automated assessment using Gemini-3-Flash (Google,

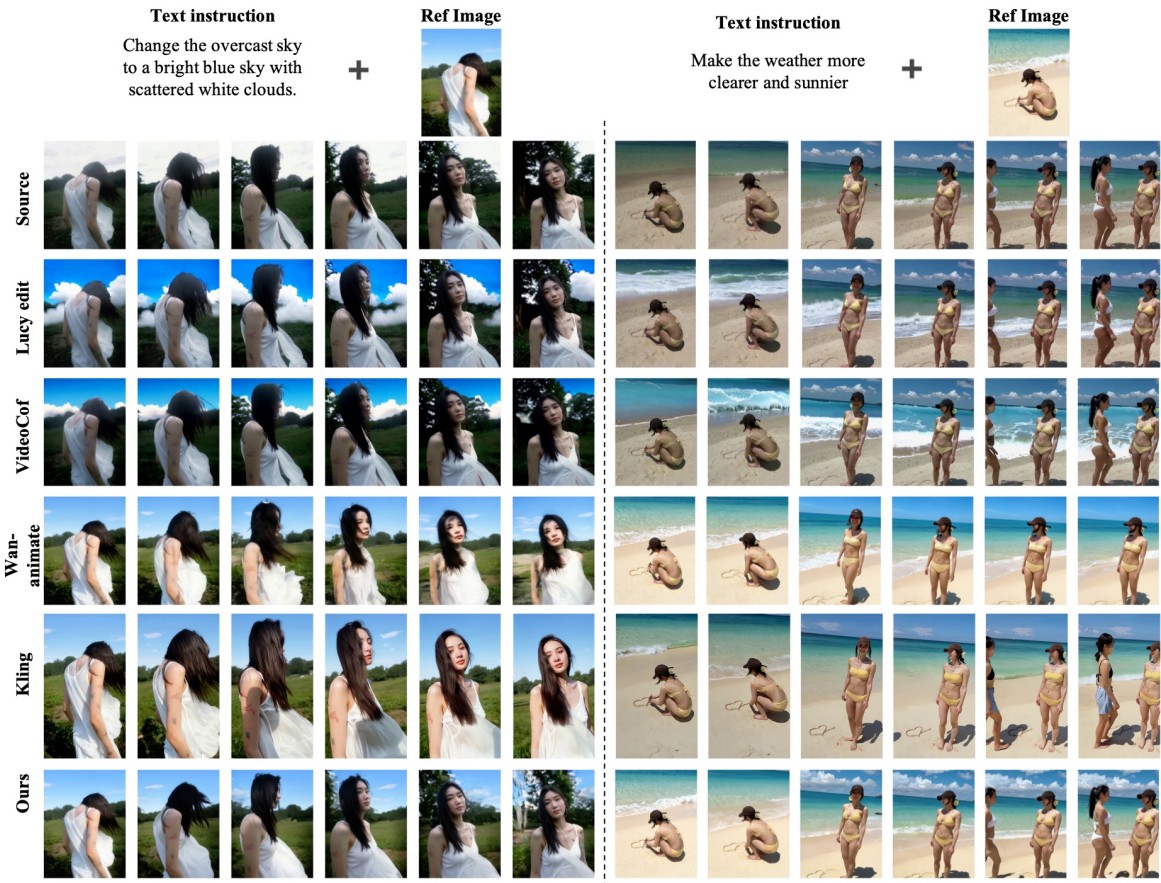

*Figure 5.* **Qualitative comparison on the complex-scenario benchmark.** In complex scenarios involving rapid motion and intricate transitions (e.g., hair color change, dramatic lighting), our model exhibits superior temporal stability and identity preservation compared to Wan-Animate, Kling O1, LucyEdit, and VideoCof. See Supplementary Videos[1] for details.

*Table 2.* Quantitative comparison across Simple and Complex scenarios. We report VLM-based evaluations and human evaluation results.

| Method | Ref. | VLM Evaluations↑ | | | | | | User Study ↑ |
|---|---|---|---|---|---|---|---|---|
| | | IA | CC | TS | PR | VA | SC | |
| *Simple (Approximate masks of modification areas are available)* | | | | | | | | |
| VACE | ✓ | 7.06 | 7.12 | 6.45 | 6.32 | 6.39 | 7.02 | 2.67 |
| LucyEdit | ✗ | 6.14 | 7.56 | 7.55 | 7.18 | 5.96 | 7.13 | 1.58 |
| VideoCof | ✗ | 7.53 | 8.04 | 8.62 | 6.62 | 6.41 | 8.28 | 1.46 |
| Kling O1 | ✓ | 8.48 | **9.03** | **8.91** | **8.68** | 8.51 | 9.31 | 3.69 |
| **Ours** | ✓ | **9.30** | 8.65 | 8.81 | 8.08 | **8.83** | **9.46** | **4.18** |
| *Complex (Masks of modification areas are impossible to acquire)* | | | | | | | | |
| LucyEdit | ✗ | 7.22 | 7.02 | 6.36 | 6.07 | 5.57 | 7.05 | 1.78 |
| VideoCof | ✗ | 7.38 | 6.66 | 6.84 | 5.57 | 4.99 | 6.34 | 1.47 |
| Wan-Animate | ✓ | 8.87 | 7.78 | 7.83 | 7.47 | 7.73 | 8.98 | 3.03 |
| Kling O1 | ✓ | 8.68 | 7.71 | 8.11 | 7.78 | 7.74 | 9.14 | 3.61 |
| **Ours** | ✓ | **9.23** | **8.05** | **8.27** | **8.18** | **8.09** | **9.22** | **3.75** |

2026) across six dimensions—Instruction Adherence (IA), Content Consistency (CC), Temporal Smoothness (TS), Physical Realism (PR), Visual Aesthetics (VA), and Style Coherence (SC)—each scored on a 0–10 scale (see Appendix C for detailed scoring criteria); (2) a user study with

30 participants providing holistic ratings on a 1–5 scale based on instruction adherence, video quality, and preservation of non-edited regions, serving to verify that VLM evaluations align with human preferences. To address potential bias between our backbone (Qwen3-VL) and evaluator (Gemini-3-Flash), we additionally replicate the automated evaluation with InternVL3.5-8B (Wang et al., 2025b) as an independent open-source evaluator (Appendix C.5), and report inter-rater reliability and paired Wilcoxon significance tests for the human study (Appendix D). We do not report SSIM/LPIPS, as these metrics measure only structural similarity and are ill-suited for editing tasks where the generated video is expected to differ substantially from the input (see Appendix E for detailed analysis).

**Quantitative Results.** Table 2 reports results across both subsets. On the *Simple Subset*, our method leads in IA (9.30), VA (8.83), and SC (9.46), while ranking second on CC, TS, and PR. Nevertheless, our user study indicates that human preference still favors our method overall. We attribute this discrepancy between VLM-based metrics and human judgment to the relatively straightforward nature of

this subset, which features slow motion and unambiguous editing objectives. On the *Complex Subset*, the advantages of our method become more pronounced: it achieves the highest scores across all six VLM evaluation dimensions and ranks first in human preference, outperforming both the open-source state-of-the-art Wan-Animate and the commercial system Kling O1.

**Qualitative Results.** We present visual comparisons in Figures 4 and 5. On the Simple Subset, our method faithfully executes object addition, removal, and replacement while managing global illumination—producing realistic shadows and maintaining background consistency. On the Complex Subset, our model demonstrates superior stability under challenging scene transitions. Notably, when objects are occluded in the reference, our method preserves facial identities more reliably than WanAnimate and Kling O1, and adheres to editing instructions more faithfully than LucyEdit and VideoCof.

*Table 3.* Ablation on conditioning architecture.

| Architecture | IA ↑ | CC ↑ | TS ↑ | PR ↑ | VA ↑ | SC ↑ |
|---|---|---|---|---|---|---|
| Decoupled Enc. + Dual Cross-Attn | 6.76 | 6.10 | 5.88 | 5.92 | 5.87 | 7.45 |
| Unified Enc. + Dual Cross-Attn | 8.51 | **8.24** | 7.68 | 7.88 | 7.42 | 8.03 |
| Unified Enc. + Fused Cross-Attn | 8.53 | 8.22 | 7.87 | 7.91 | 8.08 | 9.00 |
| **Unified Enc. + Self-Attn (Ours)** | **9.23** | 8.05 | **8.27** | **8.18** | **8.09** | **9.22** |

*Table 4.* Ablation on VLM layer selection.

| Layer Selection | IA ↑ | CC ↑ | TS ↑ | PR ↑ | VA ↑ | SC ↑ |
|---|---|---|---|---|---|---|
| First Layers Only | 8.03 | 7.16 | 6.85 | 6.98 | 6.65 | 8.44 |
| Last Layer Only | 9.04 | **8.28** | 8.03 | 7.89 | **8.11** | 9.11 |
| **First + Last Layers** | **9.23** | 8.05 | **8.27** | **8.18** | 8.09 | **9.22** |

## 6.3. Ablation Study

We conduct systematic ablation studies to validate our architectural decisions. All variants are trained under identical settings (Section 6.1). We evaluate exclusively on the Complex Subset for two reasons: (1) it demands joint semantic reasoning and spatial precision, directly testing our core design goals; (2) its challenging conditions (e.g., scene transitions, occlusions) amplify performance differences between architectures that may be masked in simpler scenarios.

### 6.3.1. CONDITIONING ARCHITECTURE

To validate unified self-attention for integrating multiscale VLM features, we compare four architectures:

- **Decoupled Encoder Dual Cross-Attention.** T5 encodes text and CLIP vision module encodes the reference frame via separate cross-attention branches, following (Wang et al., 2025a).

- **Unified Encoder Dual Cross-Attention.** First and last layer VLM features ($\tilde{\phi}_1, \tilde{\phi}_L$) are injected through dedicated cross-attention layers.
- **Unified Encoder Fused Cross-Attention.** Multiscale features are concatenated, projected via the adapter (Section 4.2), and injected through a single cross-attention branch.
- **Unified Encoder Self-Attention (Ours).** Conditional tokens **c** are concatenated with visual tokens and processed jointly through self-attention.

As shown in Table 3, our Unified Encoder Self-Attention architecture achieves superior performance across most metrics. The substantial gap between decoupled and unified encoders (row 1 vs. rows 2–4) confirms the benefit of joint multimodal encoding. Among unified encoder variants, self-attention outperforms cross-attention alternatives, suggesting that processing conditional and visual tokens within a shared latent space enables more effective interaction. Qualitative comparisons are provided in Appendix F.

### 6.3.2. LAYER SELECTION FOR VLM FEATURE EXTRACTION

As discussed in Section 4.2, we extract features from the first and last layers of Qwen3-VL to capture complementary information. We compare against two alternatives:

- **First Layers Only**, emphasizing low-level spatial structures
- **Last Layer Only**, focusing on high-level semantic context

As shown in Table 4, First Layers Only achieves reasonable semantic alignment (IA: 8.03) but suffers in temporal and spatial metrics (TS: 6.85, PR: 6.98), indicating that low-level features alone struggle to maintain video coherence. Last Layer Only excels in content consistency (CC: 8.28) but shows slightly weaker spatial precision than our combined design. By leveraging both layers, our First + Last configuration achieves the best overall performance on IA, TS, PR, and SC, effectively combining semantic depth with spatial fidelity.

## 7. Conclusion

We presented MiVE, a reference-guided video editing framework that leverages multiscale VLM features within a unified self-attention architecture. Motivated by the observation that early and final VLM layers provide complementary spatial and semantic cues, MiVE integrates these features with reference-aware video latents in a shared attention space. Experiments and ablations show that this design improves edit fidelity, temporal consistency, and preservation of unedited content across diverse and challenging editing scenarios, especially under complex motion and occlusion.

# Acknowledgement

This work was supported in part by the Fundamental and Interdisciplinary Disciplines Breakthrough Plan of the Ministry of Education of China (No. JYB2025XDXM504) and the National Natural Science Foundation of China (Nos. 62576187).

# Impact Statement

This work studies reference-guided video editing, which aims to propagate visual edits from a reference image throughout a video sequence while preserving original motion and unedited content. The proposed method advances the fidelity and consistency of video editing without requiring explicit masks, which may benefit applications such as film post-production, content creation, advertising, and visual effects.

At the same time, like other video generation and editing techniques, reference-guided video editing could potentially be misused to alter visual content in misleading or deceptive ways, such as creating deepfakes or manipulating video evidence. This work is intended for research purposes, and we encourage responsible use in accordance with existing ethical guidelines for generative models and visual media editing. We advocate for the development of detection mechanisms and watermarking techniques to mitigate potential misuse.

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

# Appendix

## A. Algorithmic Details of the Diagnostic Framework

To facilitate reproducibility, we provide the formal procedure for the Cross-Modal Attention Diagnostic Extraction used in our portrait video editing analysis. This process identifies how semantic information from textual prompts interacts with visual features across different VLM layers.

---

**Algorithm 1** Cross-Modal Attention Diagnostic Extraction for Portrait Video

---

1: **Input:** Unified context sequence $\mathbf{C} \in \mathbb{R}^{L \times d}$, total layers $L_{total}$, text index set $\mathcal{I}_{txt}$ ($|\mathcal{I}_{txt}| = N$), visual index set $\mathcal{I}_{vis}$ ($|\mathcal{I}_{vis}| = M$).
2: **Output:** Layer-wise cross-modal attention sub-matrices $\{\mathbf{A}_{txt \to vis}^{(l)}\}_{l=1}^{L_{total}}$.
3:
4: Initialize hidden state $\mathbf{H}^{(0)} \leftarrow \mathbf{C}$.
5: **for** $l = 1$ **to** $L_{total}$ **do**
6:    {// Project hidden states to Query and Key manifolds}
7:    $\mathbf{Q}^{(l)} \leftarrow \text{Norm}_q(\text{Reshape}(\mathbf{H}^{(l)}\mathbf{W}_q^{(l)}))$.
8:    $\mathbf{K}^{(l)} \leftarrow \text{Norm}_k(\text{Reshape}(\mathbf{H}^{(l)}\mathbf{W}_k^{(l)}))$.
9:
10:    {// Compute multi-head attention and average across $h$ heads}
11:    $\bar{\mathbf{A}}^{(l)} \leftarrow \frac{1}{h}\sum_{i=1}^{h} \text{Softmax}\left(\frac{\mathbf{Q}_i^{(l)}(\mathbf{K}_i^{(l)})^\top}{\sqrt{d_k}}\right) \in \mathbb{R}^{L \times L}$.
12:
13:    {// Extract cross-modal interaction sub-matrix for portrait cues}
14:    $\mathbf{A}_{txt \to vis}^{(l)} \leftarrow \bar{\mathbf{A}}^{(l)}[\mathcal{I}_{txt}, \mathcal{I}_{vis}] \in \mathbb{R}^{N \times M}$.
15:
16:    {// Advance to the next transformer block}
17:    $\mathbf{H}^{(l)} \leftarrow \text{TransformerLayer}_l(\mathbf{H}^{(l-1)})$.
18: **end for**
19: **return** $\{\mathbf{A}_{txt \to vis}^{(l)}\}_{l=1}^{L_{total}}$.

---

## B. Data Filtering Pipeline

We employ Qwen3-VL-8B as an automated quality judge to filter training pairs. Each sample is evaluated by comparing the source and edited videos against the editing instructions. The evaluation follows a three-stage protocol with decimal scoring (0.0–10.0), where only samples scoring $\geq 8.5$ are retained.

### B.1. Evaluation Protocol

**Stage 1: Hard Rejection (Score $< 5.0$).** Samples are immediately rejected if any of the following criteria are violated:

- **Aesthetic Failure**: Poor image quality, blurriness, distortion, color anomalies, visual artifacts, or frame corruption.
- **Facial Distortion**: Asymmetric eyes, twisted facial features, deformed faces, or unnatural expressions.
- **Static Content**: Videos where $>50\%$ of frames remain essentially static, lacking meaningful motion.
- **Physical Implausibility**: Floating objects, gravity-defying fluid dynamics, or clothing that does not respond naturally to movement.
- **Instruction Failure**: The edit is imperceptible or not executed.

**Stage 2: Quality Threshold (Score $5.0$–$8.4$).** Samples without hard failures but exhibiting minor flaws are rejected:

- **Score 5.0–7.5**: Technically correct but *not aesthetically pleasing*—e.g., slight color disharmony, visible edge artifacts, or unnatural blending in local edits.
- **Score 7.6–8.4**: Acceptable quality but *lacking refinement*—e.g., hair remaining static while the body moves, insufficient motion dynamics, or imperfect detail handling.

**Stage 3: Retention (Score $\geq 8.5$).** Only samples meeting all of the following criteria are retained:

- **Aesthetic Excellence**: Visually appealing, natural texture (or refined stylization for global edits), harmonious colors.
- **Seamless Integration** (local edits): Edit boundaries are invisible; lighting, noise, and texture match the original.
- **Rich Motion**: Noticeable and natural dynamic changes (character movement, camera motion, object dynamics).
- **Physical Plausibility**: Realistic weight, inertia, and material behavior.
- **Instruction Fidelity**: The edit faithfully executes the instruction while maintaining visual coherence.
- **Facial Quality** (if applicable): Natural, well-proportioned faces with vivid expressions.

### B.2. Edit-Type-Specific Criteria

For **global edits** (e.g., style transfer), a degree of stylization or "AI aesthetic" is acceptable, provided the result remains visually appealing. However, facial consistency is strictly enforced: faces in the edited video must closely match those in the source (similar features, expressions, and angles), except in explicitly stylized or animated scenarios.

For **local edits**, strict realism is required. The edited region must blend seamlessly with the surrounding environment in terms of lighting direction, color temperature, material texture, and physical behavior. Any visible "pasted-on" appearance results in rejection.

## B.3. Prompt Template

The complete evaluation prompt is provided below:

> **Quality Evaluation Prompt**
>
> **System**: You are a video quality expert with extremely high aesthetic standards. Compare the source and edited videos, and strictly judge whether the edit "{instruction}" was executed with high quality.
> Use decimal scoring (one decimal place, e.g., 7.5, 8.2).
> **Stage 1 - Hard Rejection** ($<5.0$): Aesthetic failure, facial distortion, static content, physical implausibility, or instruction failure.
> **Stage 2 - Quality Threshold** (5.0–8.4): Minor flaws in aesthetics, blending, motion, or detail refinement.
> **Stage 3 - Retention** ($\geq 8.5$): Aesthetic excellence, seamless integration, rich motion, physical plausibility, instruction fidelity, and facial quality.
> **Output Format** (strictly three lines):
> Score: [1.0–10.0]
> Retain: [Yes/No] (Yes only if score $\geq$ 8.5)
> Reason: [Concise critique covering aesthetics, motion, and integration]

# C. VLM-based Evaluation Protocol

We employ Gemini-3-Flash-preview as an automated evaluator to assess video editing quality across six complementary dimensions. For each evaluation, we uniformly sample 40 frames from both the source and edited videos and provide them to the VLM along with the editing instruction and (optionally) a reference image.

## C.1. Evaluation Dimensions

**Instruction Adherence (IA).** Measures whether the edit faithfully executes the given instruction. The evaluator identifies differences between source and edited videos, compares them against the expected outcome, and checks for completeness of required modifications (e.g., synchronized shadows, reflections, color changes).

**Content Consistency (CC).** Assesses whether regions *not* targeted by the instruction remain unchanged. This includes stability of non-target objects, backgrounds, lighting, and motion trajectories. Unintended modifications to non-target areas are penalized.

**Temporal Smoothness (TS).** Evaluates frame-to-frame continuity and motion coherence. Flickering, jittering, texture jumps, abrupt shape changes, or inconsistent motion speed/direction are penalized.

**Physical Realism (PR).** Judges whether the edited video adheres to physical and biological laws, including realistic motion dynamics (inertia, gravity, acceleration), plausible collisions and occlusions, and consistent lighting/shadow behavior.

**Visual Aesthetics (VA).** Evaluates the visual appeal of the edited regions, including composition, color harmony, texture clarity, and overall visual quality. Artifacts, blurriness, resolution drops, and rough edges are penalized.

**Style Coherence (SC).** Measures the consistency of color grading, lighting direction, texture style, and visual atmosphere across the entire video. Local style breakdowns or inconsistent transitions are penalized.

## C.2. Scoring Criteria

Each dimension is scored on a 0–10 scale with the following guidelines:

- **9.5–10**: Flawless—no visible defects.
- **8.0–9.4**: Minor imperfections—core requirements met with slight deviations.
- **6.0–7.9**: Moderate issues—noticeable defects but main content remains coherent.
- **4.0–5.9**: Significant problems—major requirements unfulfilled or erroneous.
- **0.0–3.9**: Severe failure—completely inconsistent with requirements.

Additionally, a global rule enforces that if the source and edited videos show negligible difference (i.e., the instruction was not executed), all dimension scores are capped at 6.0.

## C.3. Reference Image Handling

When a reference image (typically the edited first frame) is provided, the evaluator uses it as a style anchor. If an element disappears in later frames of the edited video, the evaluator cross-references the source video: if the corresponding element also disappears in the source (e.g., due to occlusion or camera motion), no penalty is applied. Only genuine style regressions (e.g., color/lighting reverting to the original) are penalized.

## C.4. Output Format

For each dimension, the VLM outputs a JSON object containing the score and reasoning:

```
{
  "<DIM>_score": 8.5,
  "reasoning": "Detailed justification..."
}
```

where `<DIM>` $\in$ {IA, CC, TS, PR, VA, SC}.

## C.5. Cross-Evaluator Validation with InternVL

To address potential representational bias between our backbone (Qwen3-VL) and our evaluator (Gemini-3-Flash), we replicate the full automated evaluation using InternVL3.5-

8B (Wang et al., 2025b) as an independent, fully open-source evaluator under the identical protocol described above. Table 5 reports the mean score (averaged across the six dimensions) and the resulting rank on each subset, alongside the Gemini and human-study results.

*Table 5.* Cross-evaluator triangulation. Mean score (across six dimensions for VLM evaluators; 1–5 Likert for humans) with rank in parentheses. On the Complex Subset, all three evaluators produce identical top-3 rankings; on Simple, the only discrepancy is a marginal top-2 swap under InternVL ($\Delta = 0.07$).

| Method | Gemini | InternVL | Human |
|---|---|---|---|
| *Simple Subset* | | | |
| Ours | **8.86 (#1)** | 8.79 (#2) | **4.18 (#1)** |
| Kling O1 | 8.82 (#2) | **8.86 (#1)** | 3.69 (#2) |
| VACE | 6.73 (#5) | 8.68 (#3) | 2.67 (#3) |
| VideoCof | 7.58 (#3) | 8.38 (#4) | 1.46 (#5) |
| LucyEdit | 6.92 (#4) | 7.80 (#5) | 1.58 (#4) |
| *Complex Subset* | | | |
| Ours | **8.51 (#1)** | **8.68 (#1)** | **3.75 (#1)** |
| Kling O1 | 8.19 (#2) | 8.66 (#2) | 3.61 (#2) |
| Wan-Anim. | 8.11 (#3) | 8.63 (#3) | 3.03 (#3) |
| LucyEdit | 6.55 (#4) | 7.69 (#5) | 1.78 (#4) |
| VideoCof | 6.30 (#5) | 7.88 (#4) | 1.47 (#5) |

The convergence between two structurally different VLM evaluators (Gemini, closed-source; InternVL, open-source) and a prompt-independent human study confirms that our ranking reflects genuine quality differences rather than evaluator-specific bias.

## D. Statistical Analysis of Human Study

To characterize the reliability of the 30-participant human study and the significance of its conclusions, we report two complementary statistics.

**Inter-rater reliability.** We compute Krippendorff's $\alpha$ (ordinal) across raters: $\alpha = 0.634$ on Simple (32 raters), $\alpha = 0.506$ on Complex (28 raters), and $\alpha = 0.581$ overall (30 raters). These values are consistent with the moderate agreement typically observed in subjective video assessment on a 1–5 Likert scale, and arise from differences in personal stylistic preference rather than evaluator confusion.

**Statistical significance.** We apply paired Wilcoxon signed-rank tests on the per-video human ratings between our method and each baseline. Results are summarized in Table 6.

All comparisons except Ours vs. Kling O1 on the Complex Subset are significant at $p < 0.01$. The non-significant gap on Complex ($p = 0.247$) is consistent with the small numerical margin (3.75 vs. 3.61) reported in Table 2 and is faithfully reflected in our claims.

*Table 6.* Wilcoxon signed-rank tests (paired, two-sided) on human preferences, ours vs. each baseline.

| Comparison | Simple $p$ | Complex $p$ |
|---|---|---|
| Ours vs. Kling O1 | 0.0087** | 0.247 (n.s.) |
| Ours vs. VACE / Wan-Anim. | <0.0001*** | <0.0001*** |
| Ours vs. LucyEdit | <0.0001*** | <0.0001*** |
| Ours vs. VideoCof | <0.0001*** | <0.0001*** |

## E. Limitation Analysis of Traditional Metrics

Traditional pixel-level metrics such as SSIM and LPIPS are designed to measure perceptual similarity between images. However, in video editing tasks, they exhibit fundamental limitations. As shown in Table 7, these metrics show inconsistent trends with VLM/human evaluations across Simple and Complex subsets. The structural changes inherent in editing tasks lead to lower SSIM/LPIPS scores even for high-quality edits, while less successful edits may accidentally yield higher scores if they preserve more of the original structure. This discrepancy demonstrates that structural similarity metrics cannot reliably evaluate editing quality.

*Table 7.* SSIM and LPIPS comparison across subsets.

| Method | SSIM ↑ | | LPIPS ↓ | |
|---|---|---|---|---|
| | Simple | Complex | Simple | Complex |
| VACE | **0.920** | - | **0.069** | - |
| LucyEdit | 0.841 | **0.734** | 0.140 | **0.194** |
| VideoCof | 0.783 | 0.656 | 0.187 | 0.270 |
| Kling O1 | 0.810 | 0.381 | 0.155 | 0.479 |
| WanAnimate | - | 0.411 | - | 0.416 |
| Ours | 0.824 | 0.556 | 0.139 | 0.299 |

## F. Qualitative Ablation Results

We present visual comparisons of our architectural variants in Figure 6 and Figure 7.

**Architecture Ablation.** As shown in Figure 6, the Decoupled Encoder + Dual Cross-Attention variant struggles with conflicting effects: while it can locate editing regions from textual instructions, it fails to simultaneously preserve the original video structure and faithfully propagate the reference style. This modality gap leads to inconsistent editing quality across different scenarios.

**Layer Ablation.** As shown in Figure 7, using only the first layer features produces videos with reasonable global style but loses fine-grained local details (e.g., facial identity, texture consistency). Using only the last layer features yields reasonable semantics but the local regions do not match the reference appearance. Combining both first and last layer features (Ours) achieves the best of both worlds.

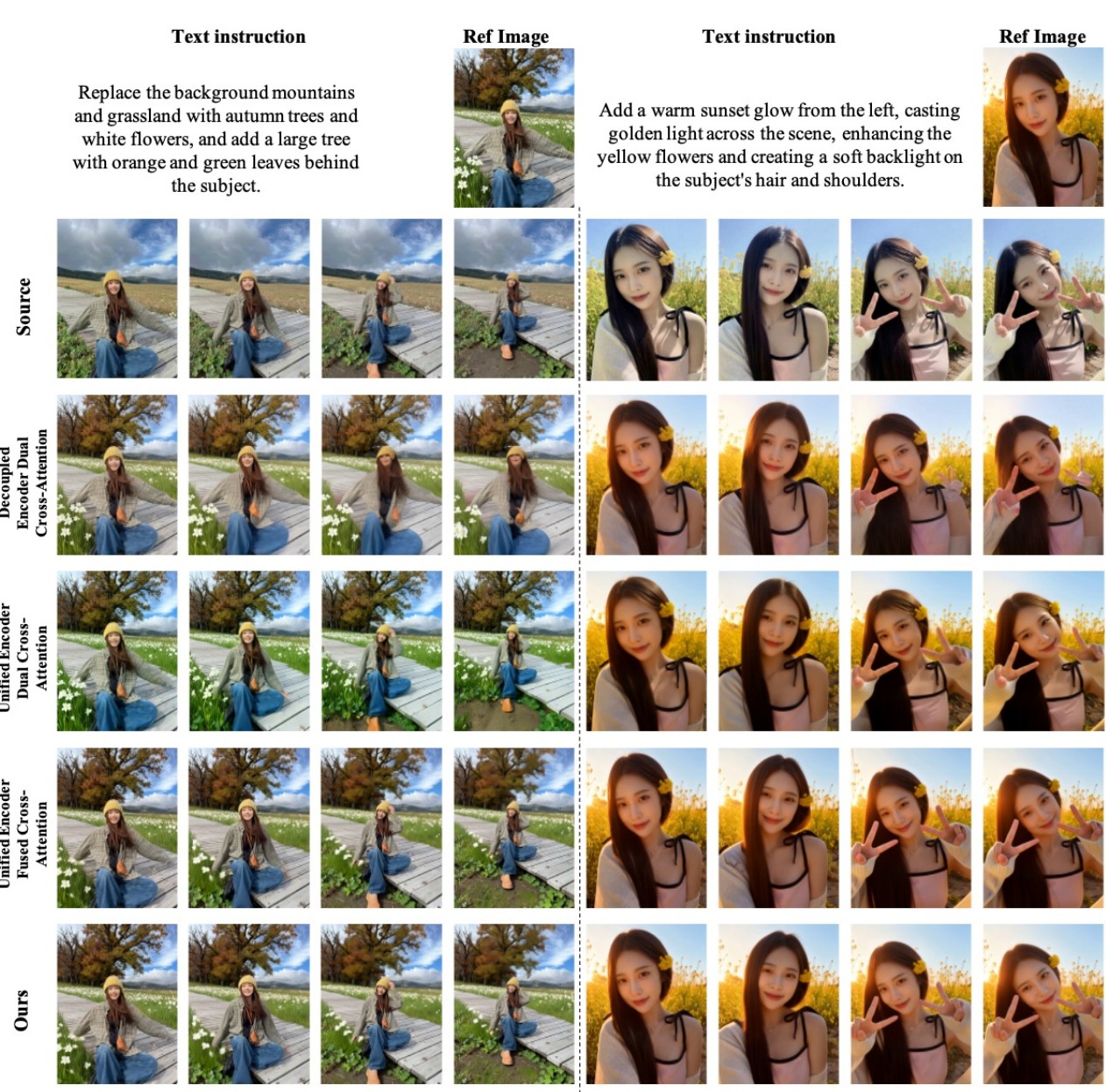

*Figure 6.* Qualitative comparison of ablation studies. Architectural variants: (1) Decoupled Enc.+Dual Cross-Attn, (2) Unified Enc.+Dual Cross-Attn, (3) Unified Enc.+Fused Cross-Attn, (4) Unified Enc.+Self-Attn (Ours).

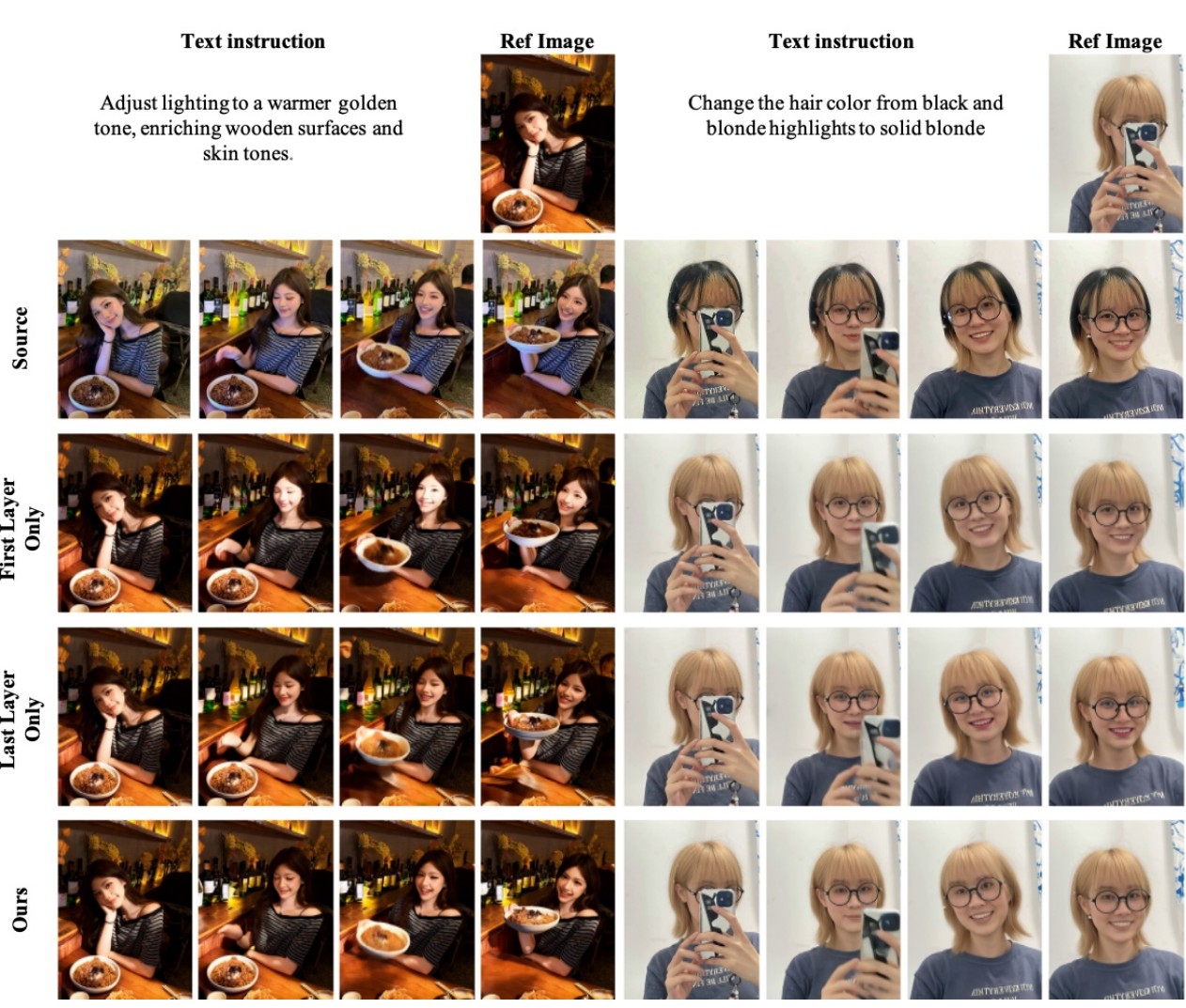

*Figure 7.* Qualitative comparison of ablation studies. (1) First layer only, (2) Last layer only, (3) First and last layers (Ours).

