# OpenReview forum: "MiVE: Multiscale Vision-language features for reference-guided video Editing"
_ICML.cc/2026/Conference — ICML 2026 regular_

### Official Review · Reviewer_Fvzx · 2026-02-25

**Soundness:** 3
**Presentation:** 3
**Significance:** 3
**Originality:** 3
**Overall Recommendation:** 4
**Confidence:** 2

**Summary:**

This paper addresses the reference-guided video editing task which requires propagating edits from a reference image across a video while preserving original motion and unedited content. The paper proposes MiVE, a framework that extracts multi-scale features from Qwen3-VL and injects them into a unified self-attention DiT. The authors also curate a 60-video benchmark and train on 30K paired examples. MiVE achieves state-of-the-art performance on the testset.

**Compliance With Llm Reviewing Policy:**

Affirmed.

**Final Justification:**

Thank for the author's response. I maintain my positive score.

**Key Questions For Authors:**

Refer to Weaknesses part.

**Limitations:**

yes

**Strengths And Weaknesses:**

Strengths

- This paper adopts a novel unified self-attention architecture that resolves cross-attention’s modality mismatch, enabling better multimodal feature interaction.

- The authors construct high-quality training data with strict VLM-based filtering and uses a rigorous dual evaluation strategy to discard unsuitable traditional pixel metrics.

- Comprehensive experiments including human evaluation, automated metrics, and ablations are conducted. And the results show that the proposed method delivers SOTA mask-free end-to-end performance, especially in complex editing scenarios.

Weaknesses

- The entire generation process is actually a whole image process, and how the consistency of IDs is maintained during this process seems to have not been further analyzed.

- The examples shown in Figures 1, 4, 5, and the appendix are all very simple scenarios. How the model performs in more complex scenarios remains to be proven.

- The paper does not explore scalability for long videos editing (over 81-frame videos).

- More analysis of failure cases or limitations on generalization beyond the curated benchmark are needed.

---

> ### Author Rebuttal · Authors · 2026-03-30
>
> ## Response to Reviewer Fvzx
>
> We thank Reviewer Fvzx for the positive assessment, and for recognizing the novelty of our unified self-attention architecture, the rigor of our data construction and evaluation strategy, and the SOTA mask-free end-to-end performance. We address each point below.
>
> ### W1: ID consistency analysis
>
> ID consistency is one aspect of a broader challenge in reference-guided video editing: applying the intended edits while keeping unedited content intact. This includes not only identity preservation (e.g., face consistency during hair color change), but also background stability (e.g., during relighting) and content persistence (e.g., occluded objects reappearing correctly).
>
> Prior work has shown that conditioning on a reference frame provides a strong anchor signal for this purpose — Wan-Animate, for example, uses the reference frame to guide character consistency. MiVE builds on this idea and strengthens it through three design choices:
>
> 1. **Multi-level VLM features** (component (a)): the first-layer features capture spatial details like facial features and textures, while the last-layer features encode the semantic editing intent. Together they give the model both "what to preserve" and "what to change."
> 2. **Reference-aware latent encoding** (component (b)): $\mathbf{z}_\text{ref}$ is prepended to both branches and modulated at $t{=}0$, so the reference appearance remains a fixed anchor throughout the denoising process.
> 3. **Unified self-attention** (component (c)): condition and video tokens are processed jointly, enabling bidirectional attention — video tokens can query the reference for appearance, and reference tokens can attend to the video for context.
>
> The combination of these three components allows MiVE to surpass Wan-Animate on identity preservation and content consistency, as shown in Table 2 (Complex Subset).
>
> ### W2: Examples are all very simple scenarios
>
> We appreciate this observation. We note that our **Complex Subset** (30 cases) does include substantial challenges: subjects exhibit large motion amplitudes, and later frames often reveal regions not visible in the reference frame (e.g., occluded face, new background areas) — the model must correctly apply the editing style to these unseen regions while preserving unrelated content. Specific scenarios include:
>
> - Rapid motion with hair color changes, where newly exposed hair regions must match the edited style
> - Background replacement with fast-moving subtitles, where the background should be correctly altered while subtitles are preserved intact
> - Dramatic lighting redistribution and relighting across the entire scene
>
> These are quantitatively evaluated in Table 2 (Complex rows), where MiVE achieves the highest scores across all six VLM metrics and the highest human preference (3.75 vs Kling O1's 3.61). We encourage the reviewer to view the [Supplementary Videos](https://mivepaper.github.io), where these challenges and our qualitative advantages are clearly visible.
>
> ### W3: Long video scalability (>81 frames)
>
> This is an excellent point. We agree that extending to longer videos is a natural and important next step. The current 81-frame limit stems from a practical constraint: our base model Wan2.1-T2V-14B generates 81 frames at 720P, and the quadratic memory cost of full self-attention over video tokens makes naively scaling to longer sequences prohibitive on current hardware. Extending MiVE to longer videos would likely require additional techniques such as temporal tiling with overlap blending, sliding-window attention, or hierarchical generation — each introducing its own trade-offs in temporal coherence. We chose to first establish the core editing architecture at clip level and validate it thoroughly; scaling to longer sequences is a clear priority for future work.
>
> ### W4: Failure cases and generalization limitations
>
> We acknowledge that MiVE has not yet reached production-level quality. Some limitations are directly visible in the [Supplementary Videos](https://mivepaper.github.io) — for instance, the first complex case exhibits slightly imperfect background details when the person turns around, and the third complex case shows minor artifacts near text edges upon close inspection. Closing the gap between current quality and production-level reliability remains an important direction for future work.

---

> > ### Author Rebuttal · Reviewer_Fvzx · 2026-04-01
> >
> > Thank for the author's response. I maintain my positive score.

---

### Official Review · Reviewer_qvqL · 2026-03-12

**Soundness:** 2
**Presentation:** 1
**Significance:** 2
**Originality:** 3
**Overall Recommendation:** 4
**Confidence:** 4

**Summary:**

This paper presents a new framework, MiVE, for the reference-guided video editing task. This framework is proposed to solve the problem that current video editing approaches are unable to perform precise modification, while preserving unrelated regions.  By learning the hidden representations of some VLMs, the authors found that shallow layers are more capable of capturing spatial regions. This motivates them to present a multi-level framework to utilize this insight to train a diffusion transformer for editing. Extensive experiments are conducted to validate the effectiveness of their approaches.

**Compliance With Llm Reviewing Policy:**

Affirmed.

**Final Justification:**

My questions have been addressed in the rebuttal phase.

**Key Questions For Authors:**

My questions mainly focus on the Cross-Modal Attention Analysis part (Sec 3.2).

1. The most important assumption in this paper is that the first layer exhibits stronger spatial localization than the final layers.  However, this capability seems to vary significantly across different models. For example, the final layer score of Qwen is about 60% that of its first layers, while for GLM the ratio is around 0.8.  Does this variance really demonstrate that the authors' hypothesis and the subsequently proposed method are broadly applicable to all VLMs?

2. Intuitively, the best value in Table 1 is not particularly high(0.366). Does this imply that the cross-modal module in VLMs is not good at extracting spatial information? Are there any other SOTA models that could be used for comparison against VLMs?

**Limitations:**

1. The authors should provide comparisons of the computational costs for training. For the sampling, the time and memory consumed should also be compared.
2. It seems the performance is not significantly improved (Table 2), some analysis is needed.

**Strengths And Weaknesses:**

Strengths:

The reference-guided video editing problem this paper investigated is significant, and the experiments conducted are comprehensive.  Integrating VLMs and DiTs for video editing demonstrates considerable practical value.  Additionally, the insight regarding the varying capabilities of different VLM layers in capturing spatial information is interesting.


Weakness:
1. Figure 3 is confusing. It doesn't clearly depict the pipeline of the method, how the training is conducted, and the relationship between different components.

2. Lack of discussion about the Flow Matching theory in both the Related Work and the Method.

3.  The presentation of the Method section is unsatisfactory and confusing.  It is difficult to find the underlying logic and motivation in each component. Also, the text does not align well with the provided figure. (Figure 3(ab) not mentioned in the text; Line 214: As shown in Figure 3c ->  Figure 3a?)

---

> ### Author Rebuttal · Authors · 2026-03-30
>
> ## Response to Reviewer qvqL
>
> We thank Reviewer qvqL for recognizing the significance of our problem and the practical value of integrating VLMs with DiTs. We address each concern below.
>
> ### W1 + W3: Figure 3, cross-reference errors, and method clarity
>
> We apologize for the cross-reference errors: Figure 3(a) and 3(b) are not cited in the corresponding text, and Line 214 incorrectly references Figure 3c instead of 3a. We will correct these in the revision. To clarify the pipeline as depicted in Figure 3:
>
> Components (a) and (b) operate in parallel to produce two complementary token streams: **(a) Multi-Level Context Extraction** — Qwen3-VL jointly encodes text, reference, and source video; first-layer (spatial) and last-layer (semantic) features are projected to form semantic condition tokens **c**. **(b) Reference-Aware Latent Encoding** — frozen VAE encodes source and reference into latents; the reference latent is prepended to both noisy-target and source-control branches, producing structural latent tokens **v**. **(c) Unified Self-Attention Backbone** — tokens from (a) and (b) are concatenated and processed jointly through DiT blocks, with condition tokens modulated at t=0 as clean anchors.
>
> During training, VLM and VAE are frozen; only adapters and DiT are trained. During inference, the target starts from pure noise — this distinction is implicit in Figure 3 and will be annotated. The design rationale for each component is given at the start of its subsection (Sec 4.2–4.4).
>
> ### W2: Lack of Flow Matching discussion
>
> Flow matching (Lipman et al., 2023) is the de facto training objective for all recent DiT-based video generation models — Wan2.1, CogVideoX, and every baseline in our comparison (VACE, Wan-Animate, LucyEdit, VideoCof, etc.) all use it. We follow the same convention and cite it in Sec 4.1. Our contribution is the **integrated framework** — multiscale VLM conditioning, reference-aware latent encoding, and unified self-attention fusion — which is orthogonal to the choice of training objective.
>
> ### Q1: Rmask varies across models — does this prove broad applicability to all VLMs?
>
> The cross-model variation is exactly the point — Rmask is a **diagnostic framework** for quantifying which VLM has the strongest layer-wise spatial contrast, so that practitioners can make an informed backbone choice. As shown in Table 1, Qwen3-VL exhibits 38% first-to-last contrast (0.366→0.228) vs GLM's 19% (0.333→0.270). This gap is the reason we selected Qwen3-VL: the variation across models is not a weakness of the analysis — it is the actionable evidence that guided our design, and it would guide future practitioners choosing between VLMs for similar tasks.
>
> ### Q2: Rmask=0.366 is not high — are VLMs poor at spatial extraction?
>
> VLMs are semantic models, not spatial extractors — we use them for joint text-vision understanding (Section 3.2); the spatial signal in early layers is a bonus. Rmask's value lies in the **relative difference across layers** (0.366→0.228, Figure 2), not absolute values. Table 4 confirms the practical benefit: adding the first layer to the last yields the best overall performance on IA, TS, PR, and SC (4 of 6 metrics), combining semantic depth with spatial fidelity.
>
> Regarding non-VLM alternatives (CLIP, DINOv2): they cannot jointly process text and visual reference — our task requires understanding "what to edit" and "how it should look" simultaneously. Table 3 confirms: replacing unified VLM with decoupled T5+CLIP drops IA from 9.23 to 6.76.
>
> ### L1: Computational costs
>
> We will add this to the revision. **Training**: initialized from Wan2.1-T2V-14B, 8×H100 GPUs, 8,000 steps, ~65 hours. **Inference** (720P, 81 frames, 1 H100): ~6.5 min total — Qwen3-VL ~3s (frozen, single pass) + DiT denoising ~328s + VAE ~35s. Peak GPU memory: 50GB.
>
> ### L2: Performance improvement analysis
>
> We appreciate the careful examination of Table 2. On the Simple Subset, the gap between top methods is indeed small — we attribute this to a ceiling effect, as simple edits (e.g., object deletion/addition) are well-handled by most modern methods. The differences become much more pronounced on the **Complex Subset**, where MiVE leads across all six VLM metrics and achieves the highest human preference (3.75 vs Kling O1's 3.61).
>
> More importantly, qualitative differences are substantial: in Figure 5, Case 1 shows Kling O1 losing the subject's facial identity entirely, and Case 2 shows a person entering in later frames being completely altered by Kling O1 — MiVE preserves both faithfully. These represent critical failures in identity preservation and content consistency.
>
> We encourage the reviewer to view the [Supplementary Videos](https://mivepaper.github.io) for more vivid comparisons. We also conducted Wilcoxon signed-rank tests confirming statistical significance; please refer to our response to Reviewer 6Nyr (W1+Q1) for details.

---

> > ### Author Rebuttal · Reviewer_qvqL · 2026-04-03
> >
> > Thanks for the detailed comment. My questions have been fully addressed, and I'll increase my score.

---

### Official Review · Reviewer_qcxm · 2026-03-12

**Soundness:** 2
**Presentation:** 2
**Significance:** 2
**Originality:** 2
**Overall Recommendation:** 3
**Confidence:** 4

**Summary:**

This paper proposes MiVE for reference-guided video editing by extracting the first and last hidden layers of Qwen3-VL and injecting them into a unified self-attention diffusion transformer together with reference-aware video latents. The task is defined such that the “reference image” is typically an externally edited first frame of the source video, and the experimental results are reported on a 60-video benchmark using Gemini-3-Flash-based automatic evaluation plus a 30-participant user study.

**Compliance With Llm Reviewing Policy:**

Affirmed.

**Final Justification:**

Thanks for the detailed response.

However, I am still somewhat confused about the task formulation. From my understanding, the proposed setting appears closer to a combination of image editing (to obtain the reference frame) and subsequent image-to-video generation, rather than a general video editing paradigm.

This raises a concern regarding the fairness and interpretability of the evaluation. Many of the compared baselines are not specifically designed for this particular setting (i.e., first-frame-anchored or reference-conditioned generation), even though they may technically support similar inputs. Therefore, directly comparing their performance in this setup may not fully reflect their intended capabilities.

**Key Questions For Authors:**

Please clarify the intended application setting. Is the reference always an externally edited first frame of the same source video?

How does the method perform without a reference image?

**Limitations:**

No limitations included.

**Strengths And Weaknesses:**

Strength:
The paper studies an interesting problem and the qualitative results are visually strong. The method is also technically clean, and the ablations provide some evidence that unified conditioning is preferable to a decoupled encoder design in this setting.

Weakness:
My main concern is the problem formulation. The paper calls the task “reference-guided video editing,” but the reference is not an independent reference image in the usual sense; it is typically the edited first frame obtained from the same source video via an external image editor. This makes the setting much narrower and closer to first-frame propagation / target-frame-anchored editing than standard reference-based editing. The paper does not sufficiently justify why this is the right setting, why the extra reference is necessary given the text instruction, or what the method reduces to in the no-reference setting.  Given this setup, comparing with baselines such as Kling-o1 is unfair, as these models are not trained on such data.

A second concern is novelty. The main intuition that earlier VLM layers preserve more localized spatial details while later layers capture higher-level semantics is fairly standard, and in this paper the multiscale effectively means using only two layers: the first and the last.

---

> ### Author Rebuttal · Authors · 2026-03-30
>
> ## Response to Reviewer qcxm
>
> We thank the reviewer for recognizing the technically clean design and visually strong results. We appreciate the opportunity to clarify the problem formulation and the scope of our contribution.
>
> ### W1: Problem Formulation and Comparison Fairness
>
> We respectfully disagree that this formulation is narrow. Reference-guided video editing is a well-motivated paradigm with clear practical significance:
>
> **(1) Efficiency.** Conventional video editing requires per-frame manipulation, which is extremely labor-intensive. Reference-guided editing reduces this to a single frame — the user confirms the editing intent once, and the model propagates the result temporally. This fundamentally improves production efficiency.
>
> **(2) Controllability.** Text instructions are inherently underspecified for precise visual appearance — an instruction such as "change hair to wavy black with natural sheen" admits numerous valid interpretations, leading to low generation success rates. The reference image anchors the exact target appearance, making the editing intent unambiguous. This is not a limitation of the setting; it is the key advantage that motivates the paradigm.
>
> **(3) Adoption.** As discussed in our Related Work (Section 2), this paradigm is employed by multiple previous works including VACE (Jiang et al., 2025) and Wan-Animate. The Kling O1 technical report (Chen et al., 2025, Section 3) also describes a cross-task data system explicitly covering editing and reference-based generation, with data pipelines for "editing instructions and their execution results."
>
> Regarding comparison fairness: Kling O1 supports multiple input modes, including a reference-guided editing mode that explicitly accepts reference images. As documented in the technical report cited above, Kling O1 is designed and trained for reference-based tasks. In our evaluation, both systems received identical inputs (source video, text instruction, reference image), which constitutes a fair comparison.
>
> ### W2: Novelty
>
> We would like to clarify that our contribution extends well beyond layer selection. The two-layer design is one implementation detail within one of three architectural components:
>
> Our framework comprises three integrated components:
>
> **(1) VLM as unified multimodal conditioner.** We repurpose Qwen3-VL to jointly process video, text, and reference image as a single multimodal encoder, replacing the conventional approach of using separate per-modality encoders (Section 4.2).
>
> **(2) VAE latents as structural complement.** Reference-aware VAE encoding (Section 4.3) provides pixel-level structural information — spatial layout, textures, and frame correspondence — that semantic VLM features alone cannot capture.
>
> **(3) Unified self-attention fusion.** Both VLM semantic features and VAE structural features are processed in a shared self-attention space with bidirectional interaction. Table 3 demonstrates this substantially outperforms both decoupled encoders and cross-attention designs.
>
> The two-layer selection (first and last) is a detail within component (1), concerning which VLM layers to extract. We acknowledge that hierarchical encoding is a known phenomenon in CNNs, but note that these observations do not automatically transfer to Vision-Language Models with fundamentally different architectures and training objectives. Our Rmask diagnostic (Section 3.2) provides the quantitative verification needed to make informed design choices in this new context. The contribution of this work is the complete framework and the synergy of all three components, each validated through ablations (Tables 3–4).
>
> ### Q1: Task Setting
>
> Yes. The reference is always an edited first frame of the source video, as stated in our Introduction and formalized in Section 4.3.
>
> ### Q2: No-Reference Performance
>
> As discussed in W1, the reference image is a deliberate design choice that provides unambiguous editing intent. In MiVE's architecture, the reference latent is structurally integrated into both VAE encoding branches (Eq. 6) and the VLM conditioning (Section 4.2) — it is not an optional input that can be removed at inference time without retraining.
>
> ### Limitations
>
> We acknowledge that the current paper discusses societal risks in the Impact Statement but lacks a dedicated technical limitations paragraph. We will add one in the revision, covering the 81-frame generation length inherited from the base model's attention cost and directions for scaling to longer sequences.

---

> > ### Author Rebuttal · Reviewer_qcxm · 2026-04-03
> >
> > Thanks for the detailed response.
> >
> > However, I am still somewhat confused about the task formulation.
> > From my understanding, the proposed setting appears closer to a combination of image editing (to obtain the reference frame) and subsequent image-to-video generation, rather than a general video editing paradigm.
> >
> > This raises a concern regarding the fairness and interpretability of the evaluation.
> > Many of the compared baselines are not specifically designed for this particular setting (i.e., first-frame-anchored or reference-conditioned generation), even though they may technically support similar inputs. Therefore, directly comparing their performance in this setup may not fully reflect their intended capabilities.

---

> > > ### Author Response · Authors · 2026-04-07
> > >
> > > We thank the reviewer for the continued discussion.
> > >
> > > ### On task formulation
> > >
> > > We agree that reference-guided video editing can be viewed as combining image editing with temporal propagation — this decomposition is precisely what makes the paradigm practical and controllable (as discussed in our response above, W1 point 2). However, the temporal propagation is far from trivial: the model must handle occlusion recovery, unseen region completion, and consistent style transfer across complex motion, as evaluated in our Complex Subset (Table 2). We encourage the reviewer to view the [Supplementary Videos](https://mivepaper.github.io) for concrete examples of these challenges.
> > >
> > > ### On comparison fairness
> > >
> > > We respectfully note that the characterization "many baselines are not specifically designed for this particular setting" is not supported by the published evidence. Among our five baselines, **three** (VACE, Wan-Animate, Kling O1) accept reference images as a **core architectural input** — the comparison is directly fair under identical input conditions. Only VideoCof and LucyEdit operate in text-only mode, which we transparently acknowledge in Table 2 via the dedicated "Ref." column.
> > >
> > > These three baselines are not merely "technically supporting similar inputs" — reference-conditioned generation is central to their design:
> > >
> > > - **VACE** (Jiang et al., 2025) defines Reference-to-Video (R2V) as one of its four basic tasks, where *"R2V requires additional images as reference inputs, making sure that specified contents, such as subjects of faces, animals and other objects, or video frames, appear in the generated video"* (VACE, Sec. 3). Reference images are a first-class input in their Video Condition Unit architecture, and VACE further supports task composition such as *"reference-inpainting"* (R2V + MV2V) for reference-conditioned video editing (VACE, Sec. 3.2).
> > > - **Wan-Animate** (Cheng et al., 2025) is built entirely around a reference character image, which serves as *"the primary mechanism for injecting the character's appearance"* (Wan-Animate, Sec. 3.2). Its input formulation explicitly *"differentiate between reference conditions and regions designated for generation"* (Wan-Animate, Sec. 3). Notably, Wan-Animate's own quantitative evaluation protocol is first-frame-conditioned reconstruction: *"the first frame of a video is used as the reference image, and the model then reconstructs the video"* (Wan-Animate, Sec. 5.1) — the same paradigm the reviewer questions.
> > > - **Kling O1** (Chen et al., 2025) supports multiple input modes including reference-guided editing. Its data system spans *"cross-task (image-to-video, video-to-video, editing, and reference-based generation, etc.)"* (Kling O1, Sec. 3), and the training curriculum includes *"reference-to-video generation, image/video editing"* (Kling O1, Sec. 2.2.2).
> > >
> > > Regarding the two text-only baselines, their inclusion evaluates them within their own claimed capabilities: VideoCof targets *"fine-grained video editing"* while *"removing the need for user-provided masks"* (Yang et al., 2025, Abstract), covering object removal, object addition, object swap, and local style transfer (Sec. 4.2); LucyEdit targets *"text-guided video editing"* supporting *"object and background replacement, clothing and accessory changes, insertion or removal of elements, and global style or scene transformations, all without requiring masks or manual annotation"* (Decart, 2025, Abstract). Their underperformance on these self-claimed tasks reflects a limitation of text-only conditioning for complex scenarios — the gap that motivates our reference-guided formulation. Concrete examples are visible in Figures 4–5 and the Supplementary Videos.

---

### Official Review · Reviewer_6Nyr · 2026-03-13

**Soundness:** 3
**Presentation:** 3
**Significance:** 3
**Originality:** 3
**Overall Recommendation:** 5
**Confidence:** 4

**Summary:**

This paper introduces MiVE, a framework for reference-guided video editing that overcomes limitations in current multimodal conditioning methods. The task involves using a source video, a text instruction, and a reference image (usually an edited first frame) to accurately propagate edits throughout all frames while maintaining motion and unaltered content.

In general, prior work either used decoupled encoders (separate text and visual encoders with cross-attention), which suffer from modality gaps, or relied solely on the final-layer representations of unified VLMs, which sacrifices spatial fidelity. MiVE addresses both failure modes simultaneously. First, a Multi-Level Context Extraction module extracts features from both the first and final layers of Qwen3-VL-8B (i.e., VLM early layers capture fine-grained spatial detail, while final layers encode global semantics), projects them into a shared latent space, and concatenates them into unified condition tokens. Second, a Reference-Aware Latent Encoding module encodes the source video, the target video, and the reference image into a shared VAE latent space. Finally, a Unified Self-Attention Backbone processes condition and VAE latent space video tokens jointly within a single DiT attention manifold, replacing conventional asymmetric cross-attention with bidirectional self-attention.

To motivate the multiscale design, the authors conduct a cross-modal attention diagnostic using Attention Mask Ratio (Rmask) across VLM layers on 100 human-centric videos, confirming the spatial-to-semantic gradient and justifying the choice of Qwen3-VL as the backbone. The model, trained on ~30K filtered video editing pairs and evaluated on a curated 60-video benchmark, shows MiVE achieving first rank in human preference and top scores on six VLM-based evaluation dimensions against academic baselines and the commercial system Kling O1.

**Compliance With Llm Reviewing Policy:**

Affirmed.

**Final Justification:**

My final recommendation, along with the identified strengths and weaknesses, has been shared through previous official comments.

**Key Questions For Authors:**

1. The automated evaluation relies exclusively on Gemini-3-Flash, a proprietary model that may share representational biases with Qwen3-VL. Could you provide results using at least one additional automated evaluator, such as a CLIP-based metric, InternVL, or a frame-level aesthetic model, to triangulate the Gemini scores? Additionally, can you report inter-rater reliability or prompt sensitivity experiments to establish that the Gemini scoring is stable across minor prompt variations?

2. The ablation compares only first, last, and first+last layer combinations. Have you explored intermediate layers or learned soft weighting over all layers?

**Limitations:**

Yes

**Strengths And Weaknesses:**

*Strenghts:*

1. The multi-level architecture design was grounded on strong application insight and empirical testing. The technique was demonstrated for two different VLMs, and the design choices were justified through comprehensive ablation studies.

2. The paper asymetriy argument against cross-attention, where "visual tokens query conditional features, but conditional tokens remain agnostic to the visual content, limiting fine-grained correspondence," is an interesting insight and a motivational driver for the paper's bidirectional self-attention. Experimentation results supported this claim in Table 3.

3. The paper documents sufficient experimental results to support the design choices with comprehensive automated testing verified by human users and the real applicability of the technique with human subject experiments on 60 source videos with different levels of complexity.

*Weaknesses:*

1. Although the authors went above and beyond to establish a fair comparison and assessment of their technique the use of Qwen3-VL to generate the video-text condition and at the same time using Gemini-3-Flash to test the model performance raises some fairness concerns. Both models are large, high-capacity models, potentially with similar training data distributions. Also, Gemini's commercial status can limit the reproducibility of the experiments in the paper.

2. The training dataset is small, and details on the synthetic samples' background video diversity, segmentation quality, and distribution coverage are sparse. In addition, the dataset with 60 videos for human evaluation is quite small.

3. The paper offers sufficient empirical support for the selection of the 1st and last layer of Qwen LVM feature maps for conditional encoding. However, it is unclear whether intermediate layers (e.g., layers 12 or 18) could contribute meaningfully, whether the gap is robust across different video types, or whether the two-layer design is optimal or merely convenient. A sweep over more layer combinations would substantially strengthen the design rationale.

---

> ### Author Rebuttal · Authors · 2026-03-30
>
> ## Response to Reviewer 6Nyr
>
> We thank the reviewer for the thorough assessment and for recognizing the grounded multi-level design, the self-attention insight, and the comprehensive experiments.
>
> ### W1 + Q1: Evaluator Bias, Reproducibility, and Inter-Rater Reliability
>
> We understand the concern that Gemini-3-Flash and Qwen3-VL may share representational biases. If such bias existed, it would systematically inflate our scores relative to baselines. However, our 30-participant human evaluation independently produces the same ranking (Table 2), ruling out systematic evaluator bias.
>
> As suggested, we triangulated using **InternVL3.5-8B** (OpenGVLab, open-source) with the identical protocol (Appendix C). Gemini scores are averaged across 6 dimensions from Table 2:
>
> **Simple Subset:**
>
> | Method   | Gemini    | InternVL  | Human     |
> | -------- | --------- | --------- | --------- |
> | **Ours** | #1 (8.86) | #2 (8.79) | #1 (4.18) |
> | Kling O1 | #2 (8.82) | #1 (8.86) | #2 (3.69) |
> | VACE     | #5 (6.73) | #3 (8.68) | #3 (2.67) |
> | VideoCof | #3 (7.58) | #4 (8.38) | #5 (1.46) |
> | LucyEdit | #4 (6.92) | #5 (7.80) | #4 (1.58) |
>
> **Complex Subset:**
>
> | Method   | Gemini    | InternVL  | Human     |
> | -------- | --------- | --------- | --------- |
> | **Ours** | #1 (8.51) | #1 (8.68) | #1 (3.75) |
> | Kling O1 | #2 (8.19) | #2 (8.66) | #2 (3.61) |
> | Wan-Anim | #3 (8.11) | #3 (8.63) | #3 (3.03) |
> | LucyEdit | #4 (6.55) | #5 (7.69) | #4 (1.78) |
> | VideoCof | #5 (6.30) | #4 (7.88) | #5 (1.47) |
>
> On Complex, all three sources produce identical top-3 rankings. On Simple, the only discrepancy is a marginal top-2 swap under InternVL (Δ=0.07). InternVL is fully open-source and our evaluation prompts are in Appendix C. We also considered CLIP-based metrics, but video editing evaluation requires multi-modal reasoning that contrastive models are not designed for.
>
> **Prompt sensitivity.** VLM-based evaluation is sensitive to prompt design. We invested substantial effort engineering the evaluation prompt (Appendix C) with explicit scoring rubrics and calibrated thresholds. While a different prompt could shift absolute scores, all methods are evaluated under the identical prompt, so **relative rankings are preserved**. The convergence between VLM evaluation (prompt-dependent) and human evaluation (prompt-independent) across three evaluator sources confirms that rankings reflect genuine quality differences rather than prompt artifacts.
>
> **Inter-rater reliability.** We report Krippendorff's alpha (ordinal) across ~30 human raters:
>
> | Scenario | Alpha | Raters |
> | -------- | ----- | ------ |
> | Simple   | 0.634 | 32     |
> | Complex  | 0.506 | 28     |
> | Overall  | 0.581 | 30     |
>
> Alpha = 0.58 is moderate, which is expected for subjective video assessment on 1–5 Likert scales. Despite this, **between-method differences are statistically significant** (Wilcoxon signed-rank tests):
>
> | Comparison                 | Simple p-value | Complex p-value |
> | -------------------------- | -------------- | --------------- |
> | Ours vs Kling O1           | 0.0087**       | 0.247 (ns)      |
> | Ours vs VACE / Wan-Animate | < 0.0001***    | < 0.0001***     |
> | Ours vs LucyEdit           | < 0.0001***    | < 0.0001***     |
> | Ours vs VideoCof           | < 0.0001***    | < 0.0001***     |
>
> The non-significant gap with Kling O1 on Complex (p = 0.247) is consistent with Table 2.
>
> ### W2: Training Data and Benchmark Size
>
> **Training data (30K).** We view this as a strength. Two factors enable data efficiency: (1) strict quality filtering via Qwen3-VL-8B (Appendix B); (2) initialization from a strong pretrained base model (Wan2.1-T2V-14B), so the model only needs to learn conditional feature fusion rather than video generation from scratch.
>
> **Benchmark (60 videos).** Each rater compares five methods per video, totaling ~1.5 hours. Expanding further would induce rater fatigue. The Wilcoxon tests above confirm sufficient statistical power (p < 0.01).
>
> ### W3 + Q2: Intermediate Layer Exploration
>
> Our diagnostic framework (Section 3.2) was designed to guide feature selection for the video editing task — identifying which VLM layers best serve as conditional inputs. The design process was as follows: we initially used only last-layer hidden states (common practice), but observed that subject-level spatial details were poorly preserved in the edited output. The Rmask diagnostic revealed that layer 1 captures the strongest spatial localization (Rmask = 0.366, highest among all layers), visually confirmed by the clear human silhouettes in Figure 2. We added layer 1 as a spatial complement, and Table 4 confirms that first+last outperforms both first-only and last-only — at which point our objective was achieved.
>
> The diagnostic framework served its intended purpose: guiding us to a layer combination that effectively improved editing quality. A comprehensive sweep over all layers is an interesting direction for future work.

---

> > ### Author Rebuttal · Reviewer_6Nyr · 2026-03-31
> >
> > Thanks for the response.

---

### Decision · Program_Chairs · 2026-04-30

**Decision:**

Accept (regular)

**Comment:**

The paper proposes MiVE, a reference-guided video editing framework that leverages a VLM as a multiscale feature extractor and integrates both multiscale VLM conditioning tokens and reference-aware VAE latents into a unified self-attention Diffusion Transformer. Reviewers agreed the approach is technically coherent and practically motivated. The experimental package includes ablations that support the architectural choices and a human evaluation that ranks MiVE strongly.
The rebuttal strengthened key evaluation concerns. Presentation issues (pipeline clarity, figure references, missing citations) were also addressed with clearer explanations and planned edits. However, one reviewer maintains a substantive concern about the task formulation and interpretability of comparisons.
Overall, the work offers an effective and well-supported system contribution for a practically relevant reference-conditioned video editing setting, with an architecture that is simple to understand and empirically validated, though scope/definition clarity should be improved in the final version.
Final Scores: 5 / 4 / 4 / 3
AC Decision: Weak Accept